# Low-energy electrons transform the nimorazole molecule into a radiosensitiser

Rebecca Meißner [1,2], Jaroslav Kočišek[3], Linda Feketeová [4], Juraj Fedor[3], Michal Fárník [3], Paulo Limão-Vieira [2], Eugen Illenberger[1,5] & Stephan Denifl [1]

While matter is irradiated with highly-energetic particles, it may become chemically modified. Thereby, the reactions of free low-energy electrons (LEEs) formed as secondary particles play an important role. It is unknown to what degree and by which mechanism LEEs contribute to the action of electron-affinic radiosensitisers applied in radiotherapy of hypoxic tumours. Here we show that LEEs effectively cause the reduction of the radiosensitiser nimorazole via associative electron attachment with the cross-section exceeding most of known molecules. This supports the hypothesis that nimorazole is selectively cytotoxic to tumour cells due to reduction of the molecule as prerequisite for accumulation in the cell. In contrast, dissociative electron attachment, commonly believed to be the source of chemical activity of LEEs, represents only a minor reaction channel which is further suppressed upon hydration. Our results show that LEEs may strongly contribute to the radiosensitising effect of nimorazole via associative electron attachment.

[1] Institut für Ionenphysik und Angewandte Physik and Center for Biomolecular Sciences Innsbruck, Leopold-Franzens Universität Innsbruck, Technikerstrasse 25, A-6020 Innsbruck, Austria. [2] Atomic and Molecular Collisions Laboratory, CEFITEC, Department of Physics, Universidade NOVA de Lisboa, 2829-516 Caparica, Portugal. [3] J. Heyrovský Institute of Physical Chemistry v.v.i., The Czech Academy of Sciences, Dolejškova 3, 18223 Prague, Czech Republic. [4] Université de Lyon; Université Claude Bernard Lyon1; Institut de Physique Nucléaire de Lyon, CNRS/IN2P3 UMR 5822, 69622 Villeurbanne, Cedex, France. [5] Institut für Chemie und Biochemie-Physikalische und Theoretische Chemie, Freie Universität Berlin, Takustrasse 3, 14195 Berlin, Germany. Correspondence and requests for materials should be addressed to S.D. (email: Stephan.Denifl@uibk.ac.at)

Ballistic electrons with kinetic energies below about 12 eV are formed as abundant secondary species in the interaction of high-energy quanta (particles or photons in the MeV range) with cellular components[1,2]. After their formation, they lose kinetic energy by a series of elastic or inelastic collisions before they reach some stage of solvation and become chemically inactive[3,4]. In the course of this collision sequence, they can initiate very effective chemical reactions via dissociative electron attachment (DEA), thereby generating negatively charged fragment ions and neutral radicals at electron energies significantly below the corresponding bond dissociation energy[5]. DEA can be a highly selective process in the cleavage of certain chemical bonds within a molecule[6]. This unique property compared to processes induced by high-energy radiation, is based on the resonant nature of DEA, i.e., the incoming electron with a specific low-energy forms a specific transient anion state. Since about $10^4$–$10^5$ secondary low-energy electrons are formed in biological matter per 1 MeV deposited primary quantum, the relevance of low-energy electrons for radiation damage of critical targets such as DNA was recognised[7]. Indeed, the importance of the DEA processes for the induction of strand breaks in dry plasmid DNA by low-energy electrons was shown by Sanche et al.[8] and very recently bond cleavage upon electron attachment was also observed for ribothymidine in the solution phase[9]. DEA was also suggested as the operative mechanism for the action of modified pyrimidines acting as radiosensitisers in tumour cells[10–12]. For example, 5-bromouracil (BrU) is a well-known radiosensitiser with a bromine atom attached at the C5 position of the pyrimidine ring. The DEA cross section for the formation of Br⁻ (accompanied by formation of the corresponding neutral radical) turned out to be two orders of magnitude larger than the cross section for the most abundant DEA reaction in thymine, the formation of the dehydrogenated parent anion by emission of a neutral H radical[11]. When BrU is incorporated into DNA due to its structural analogy to thymine ($CH_3$ group at C5 position) and uracil (H atom at C5 position), the radical sites in the DNA formed upon DEA may lead to strand breaks by subsequent local chemistry[13].

Here we study the prototype compound, nimorazole (NIMO), of the class of nitroimidazolic (NI) radiosensitisers. For few decades, NIs have been investigated in biochemical and oncologic studies. Those compounds have been considered to mimic the well-known oxygen effect which is absent in hypoxic cells due to their anaerobic environment[13,14]. The chemical structure of NIMO is included in Fig. 1. The compound is characterized by its NI ring interconnected to a morpholine ring by a carbon chain. NIMO turned out to be the only successful compound to overcome tumour hypoxia with reasonable side effects for patients[15] and therefore is used as standard chemical compound for pharyngeal and supra-glottic carcinoma in Danish radiotherapy centers[16].

In our study we obtain that dissociation plays a minor role when an electron attaches to NIMO. Instead, the formation of the parent radical anion by associative electron attachment to NIMO is a very efficient process, in particular, when the molecule is hydrated. In this case, the only notable fragment anion $NO_2^-$ becomes strongly quenched which is not related to a change in the probability of spontaneous electron emission (autodetachment). Our quantum chemical calculations for NIMO clustered with one or two water molecules fully support these experimental results. The calculations show that electron attachment to hydrated NIMO is stabilized by the presence of water. Ultimately, these results show that free radical anion formation by attachment of a low-energy electron could be the key process for radiosensitisation of hypoxic tumour cells by the electron-affinic NI radiosensitisers.

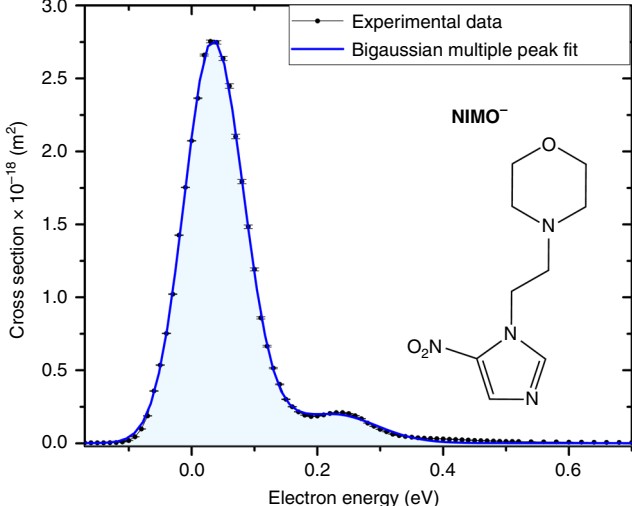

**Fig. 1** Cross-section of the nimorazole radical anion and molecular composition. The black dots (connected by the black line as guide for the eye) represent the experimentally determined cross-section values as a function of the electron kinetic energy. Statistical error margins are included for each data point and refer to the standard error of the mean, see data analysis section for details. The blue line shows a bigaussian (sum of two Gauss peaks) multiple peak fit of the experimental data. In addition, the skeletal formula of nimorazole ($C_9H_{14}N_4O_3$) with its nitroimidazole moiety and the morpholine ring is shown. Source data are provided as a Source Data file

## Results

**Formation of the NIMO radical anion upon electron attachment.** The present results indicate that *free* low-energy electrons are very effectively captured by NIMO. In contrast to halouracils, the dominant feature in electron attachment to NIMO is the very effective formation of the non-dissociated (metastable) parent anion, NIMO⁻, without the cleavage of chemical bonds. This anion is observed within a very narrow main peak close to zero eV electron energy as shown in Fig. 1. We determined the absolute cross-section of this associative attachment (AA) process to be $\sim 3 \times 10^{-18}$ m² (with an uncertainty of one order of magnitude, see "Methods" for the discussion on possible systematic errors). This cross-section is an extraordinary high value which exceeds the geometrical cross-section of the target molecule (about $0.5 \times 10^{-18}$ m²).

Attachment of free electrons is accompanied with often significant energy release comprising the kinetic energy of the incoming electron and the electron affinity of the molecule, which was calculated to be 1.31 eV (at the M062x/6-311 + G(d,p) level of theory)[17]. The anion lifetime is typically short due to electron autodetachment or molecular dissociation[5]. The formation of (metastable) parent anions with high cross-sections on a microsecond detection time scale, as for the present case, is therefore restricted to only two types of molecular systems. The first enables effective redistribution of the excess energy over the vibrational degrees of freedom such as the sulphur hexafluoride molecule $SF_6$[18] or fullerenes like $C_{60}$[19]. These two compounds are characterised by the high symmetry of the molecule (e.g., in the former case there are six equal S–F bonds), which provides favourable conditions for the formation of a metastable parent anion. The second provides an effective sink for the excess energy via intramolecular bond breakage and rearrangement[20]. Though NIMO is of low symmetry, we suppose its appreciable number of 84 vibrational degrees of freedom

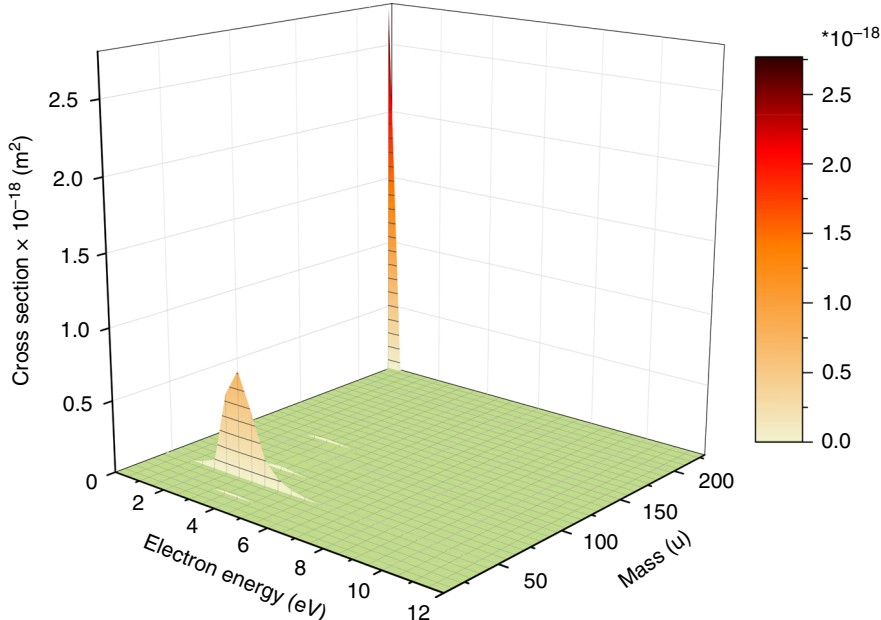

**Fig. 2** Intensity map of (fragment) anions formed upon electron attachment to nimorazole. The map shows cross-sections derived from negative ion mass spectra recorded in the interval of 0.5 eV for the electron energy range between ~0 and 12 eV. Only two abundant anions can be observed, (i) the radical anion at mass 226 u close to zero eV (see Fig. 1), and (ii) the nitrite $NO_2^-$ at mass 46 u, which is abundantly formed only in the region between 2 and 4 eV. Source data are provided as a Source Data file

provides an effective means for energy redistribution making the observation of an intense $NIMO^-$ within our observation time window of few hundred microseconds possible.

**Dissociative electron attachment to NIMO.** The only observable relevant DEA reaction is the formation of $NO_2^-$. Figure 2 shows that the cross-section for this channel is about one order of magnitude lower than that for $NIMO^-$, occurring in the electron energy range between ~2–4 eV. Other observed fragment anions have cross-sections even two orders of magnitude lower than that for the parent anion and therefore they will not be discussed here. The electron energy dependence of their cross-sections is shown in the Supplementary Figs. 1–3. The formation of $NO_2^-$ can formally be expressed as,

$$e^- + NIMO \leftrightarrow (NIMO^-)^\# \rightarrow (NIMO - NO_2) + NO_2^- \quad (1)$$

with $(NIMO^-)^\#$ representing the intermediate molecular anion. Reaction (1) represents the simple cleavage of the $C–NO_2$ bond after initial formation of a repulsive shape resonance of $\sigma^*(C–NO_2)$ character. In this case, the DEA reaction is expected as direct electronic dissociation along the repulsive C–N potential energy surface. We calculated the thermochemical threshold (free energy of reaction $\Delta G$) for this process at the M062x/6–311+G(d,p) level of theory and obtained a value of +0.53 eV (release of the intact neutral fragment, as mentioned in reaction (1)). Since we can also observe a weak $NO_2^-$ signal at electron energies close to 0 eV (see Supplementary Fig. 3), we investigated other dissociation pathways, which may lead to this anion yield. The only exothermic reaction found corresponds to the release of $C_3H_3N_2 + C_6H_{11}NO$ with $\Delta G = -0.26$ eV. The formation of $NO_2^-$ is driven by the appreciable adiabatic electron affinity (AEA) of $NO_2$ (present value 2.40 eV). However, the weak $NO_2^-$ ion yield at zero eV indicates a barrier on this reaction pathway. Therefore, autodetachment can effectively compete with the DEA channel of $(NIMO^-)^\#$ at this energy. We further note that though our calculations for the $(NIMO–NO_2)$ fragment predict an appreciable AEA of +2.07 eV, and a DEA threshold of +0.85 eV for the release of $NO_2+ (NIMO–NO_2)^-$, we

do not observe this simple reaction channel leading to the reactive nitrogen species $NO_2$[21].

**Solvation effects upon DEA to NIMO.** It is important to note that the outcome of DEA reactions may be influenced by the presence of an environment. In order to elucidate this point, we performed electron attachment experiments with hydrated NIMO, i.e. $NIMO(H_2O)_n$ clusters with $<n> \leq 14$. We chose this approach since intense research in such finite cluster systems and in the condensed phase over the last years demonstrated that in bound molecules DEA can usually still be described on a molecular site, i.e., electron attachment proceeds to an *individual* molecule which is coupled to an environment. The latter affects both, the initial attachment process and the decomposition of the intermediate molecular anion[22]. The results for clustered NIMO indicate that DEA becomes suppressed in favour of AA. The ion yield ratio of $NO_2^-/(NIMO(H_2O)_n)^-$ (which can also be interpreted as the overall ratio DEA/AA) for different solvation conditions is shown in Fig. 3. A decrease by three orders of magnitude can be observed, indicating that DEA is reduced for solvated NIMO. This behaviour is within the general trend when going from the gas phase over microsolvation into solution[23] and it is caused by parent anion stabilization via energy dissipation to the environment[24]. The detailed mechanism of the anion stabilization can be caused by (i) caging of dissociation products[25,26] and (ii) ultrafast quenching of the transient anion[27].

In view of the strong decrease of the ion yield ratio of $NO_2^-/(NIMO(H_2O)_n)^-$, one could also speculate about option (iii), a change in the autodetachment probability which leads to the decrease of the $NO_2^-$ channel. This process will lead to a change of the position and the width of the resonance with the hydration[28]. To elucidate this question, we performed electron energy dependent ion yield measurements for the $NO_2^-$ and the parent anion at different hydration conditions. The yields are shown in Fig. 4. As already discussed, in the isolated molecule the parent anion is formed at near zero energies and $NO_2^-$ at around 3 eV. With increased hydration, the intensity of the low-energy

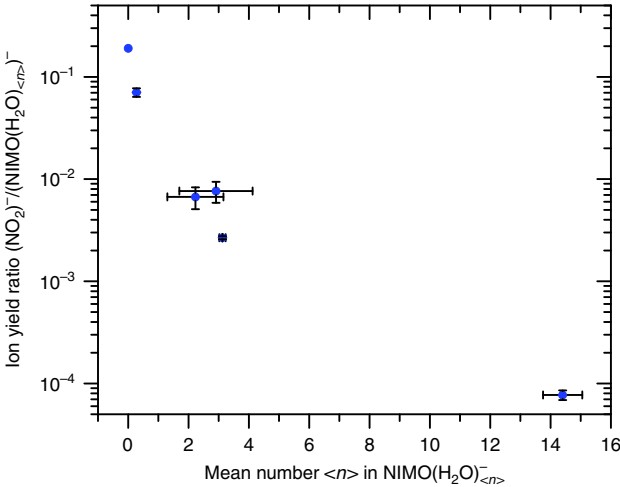

**Fig. 3** Closing of the most intense DEA channel with hydration—total ion intensity. The relative intensity of DEA to AA (blue data points) is expressed as the ratio of total yields of $NO_2^-$ to $NIMO(H_2O)_n^-$ parent ions at different hydration conditions (see Supplementary Fig. 4). The level of hydration is characterised by the mean number <n> of water molecules attached to the parent anion in $NIMO(H_2O)_n^-$ clusters. The number <n> is lower, but directly proportional to the number of water molecules in the neutral precursor cluster (see ref. [24] for details). The statistical uncertainty described by the standard deviation is shown as error bars, see data analysis section for more details. Source data are provided as a Source Data file

peak in the spectrum of the parent anion decreases. This is caused by the fact that the attachment at low electron energies results into formation of $NIMO(H_2O)_n^-$ anions instead of bare $NIMO^-$. The new feature is that when the molecule is hydrated with only few water molecules, the parent anion signal appears at around 3 eV. Therefore, the 3 eV resonance, which normally results in the dissociation, is now stabilized and the excess energy is presumably released by evaporation of water. Such evaporation then results in the formation of the parent anion. The 3 eV resonance shape weakly depends on the hydration degree: at lower hydration conditions (I, II) only the lower energy part of the resonance is stabilized, at higher hydration conditions (III, IV) the parent anion curve practically resembles that of the $NO_2^-$ curve of the isolated molecule. Finally, at the highest hydration conditions (V), the 3 eV resonance practically disappears for both $NO_2^-$ and the parent anion. Both resonances now result in the formation of larger $NIMO(H_2O)_n^-$ cluster anions. The energy acquired by the attachment process is not enough to dissociate the molecule, nor to evaporate all the water monomers from the cluster.

On the basis of the aforementioned observation, we can clearly exclude option (iii) from the explanation of the NIMO stabilisation against DEA. If mechanism (iii) is operative, one would expect only the disappearance of the 3 eV resonance but no formation of the parent anions at the same energy. The data also demonstrates the important fact that the primary electron attachment target at low energies is still the NIMO molecule, despite of the possible electron attachment to the surrounding water network[29]. This effect can be ascribed to the large de-Broglie wavelength of the impinging low-energy electron, which is about 0.7 nm at the electron energy of 3 eV and to the positive electron affinity of NIMO.

Moreover, we have also calculated structures of anionic NIMO hydrated by one and two water molecules. The anions were optimized at M062x/6–31 + G(d,p) level of theory and basis set.

The lowest energy conformers are shown in Fig. 5, while all the different binding motifs investigated are summarized in Supplementary Figs. 5 and 6. From all the structures found for the anionic NIMO hydrated by water, we can conclude that the anion is more and more stabilized with the increasing number of water molecules present. The AEAs of NIMO hydrated by one water molecule increases to 1.36–1.76 eV (see Supplementary Fig. 5), while for the NIMO hydrated by two water molecules increases to the range of 1.53–1.96 eV (Supplementary Fig. 6). Additionally, the vertical detachment energy (VDE) of the NIMO anion also increases with the increasing number of water molecules. While for the bare NIMO anion, the VDE is 1.68 eV, addition of one water molecule increases the VDE to 1.83–2.32 eV (see Supplementary Fig. 5), and the presence of two water molecules increases the VDE to 2.11–2.64 eV (Supplementary Fig. 6). Notably, for the most stable conformers of the hydrated NIMO anion in Fig. 5, the water molecules seem to cluster around the $-NO_2$ group, however, this is not the case for neutral structures. It can be seen in Supplementary Fig. 5 that the preference for water binding to neutral NIMO are, e.g., the O atom of the morpholine ring or the N atom of the imidazole ring. Nevertheless, the relative energies of these neutrals within the error of the calculation can be assumed to be isoenergetic.

## Discussion
Together with the remarkably high electron attachment cross-section, the quenching of DEA in the solvated molecule is the second key outcome of our studies. The calculations for NIMO clustered with one or two water molecules indicate that the attachment of an electron to hydrated NIMO is stabilized by the presence of water, the attraction of water molecules to the $-NO_2$ group and a fast dissipation of the excess energy due to the strong interactions between the O−H oscillators of water molecules[30,31] and their evaporation, while the increased VDE makes the autodetachment also less probable. Though very simplified systems are studied here compared to the complex cellular environment, the time evolution of radiation damage and the peculiar features of electron attachment mentioned above may allow nonetheless some predictions on the action of NIMO as radio-sensitiser in vivo. Due to the favourable adiabatic electron affinity of NIs, it was previously suggested that NIs become only active after the reduction[13,32], where the rate of the reduction determines the uptake of the radiosensitiser by the cell. Therefore, free radical anion formation by low-energy electrons seems to be the key process for radiosensitisation of hypoxic tumour cells by NIs. This conclusion is further supported by the result that NIs have to be present at the instant of radiation for a radiosensitising effect, while when adding them at a later stage (after few ms) the effect vanished[13].

After the reduction process of NIs in a cellular environment, it was suggested that the intact anion itself is not the cytotoxic species which attacks DNA. Instead, the NI radical anion may become protonated in hypoxic cells[32,33], and then the neutralised radical compound could bind to DNA. In such case, chemical reaction schemes exist, which lead to strand breaks in DNA[34,35]. These schemes rely on the binding of neutral NIs at DNA sites attacked by hydroxyl radicals. The latter radicals are formed simultaneously by radiolysis processes. Therefore, NIs were suggested to mimic the oxygen effect in hypoxic tumours.

Finally, we note that hypoxic cells without a radiosensitiser usually require two to three times higher radiation dose for cell death compared to normally-oxygenated cells[13]. Electron-affinic radiosensitisers such as NIs may significantly lower the doses and consequently the side effects of the therapy[36]. Here, we demonstrate a fundamental mechanism of their accumulation in tumour

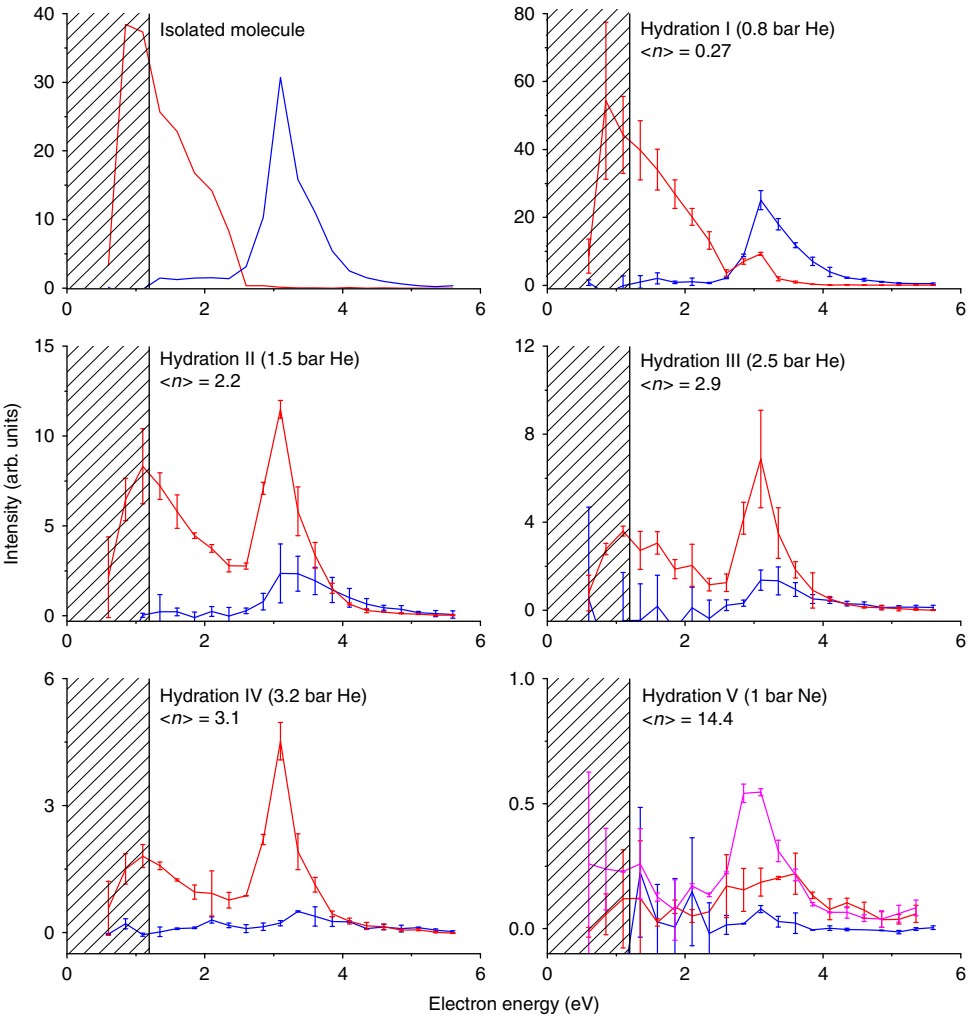

**Fig. 4** Closing of the most intense DEA channel with hydration—energy dependence. Energy dependent ion yields for the formation of $NO_2^-$ (blue) and parent $NIMO^-$ (red) ions at different hydration conditions. The level of hydration is characterised by the mean number <$n$> of water molecules attached to the parent anion in $NIMO(H_2O)_n^-$ clusters. At the highest hydration conditions, we are showing also the yield for $m/z = 280$ $NIMO(H_2O)_3^-$ anion (pink). The shape of the ion yield curve in the patterned part of the spectra (below about 1.2 eV) is influenced by the decrease of the electron current in the cluster beam setup, see ref. [40] for details. The statistical uncertainty described by the standard deviation is shown as error bars, see data analysis section for more details. Source data are provided as a Source Data file

cells, which may be used in further improvement of these important therapeutic agents.

## Methods

**Experimental design.** The present study was carried out at two different crossed electron-molecular/cluster beam setups. The single-molecule data was collected at the Wippi apparatus in Innsbruck, a detailed description can be found in ref. [37]. An oven serves as inlet for the NIMO sample. A capillary of 1 mm diameter is mounted onto it to guide the evaporated sample towards the interaction region. As ionisation source serves a hemispherical electron monochromator (HEM). It provides electrons with a narrow energy distribution (~100 meV) with Gaussian profile. The attachment processes take place in the region where molecular beam and electrons cross. Measurements at different electron energies are enabled by applying an appropriate acceleration potential in the HEM shortly before the interaction region. The negatively charged parent and fragment ions formed are subsequently extracted into a quadrupole mass analyser by a weak electrostatic field. The quadrupole has a nominal mass range of 2048 u and is utilised for mass selection. Thus, combining the HEM and the mass filter, the formation efficiency of selected fragments at varying energies can be studied. The ions are detected by a channel electron multiplier and counted by a preamplifier with analog-to-digital converter unit. The mass spectrometer is operated under high vacuum (~$10^{-8}$ mbar background pressure).

For cluster experiments, the CLUster Beam (CLUB) apparatus in Prague was used, for a detailed review refer to ref. [38]. In the present study, the configuration of

the experiment was identical to that one described in ref. [25]. For cluster production, helium or neon gas is humidified by a Pergo gas humidifier. A Nafion tubing gas line passes through a water bath and its membrane selectively permeate water vapour. The humidified gas is introduced into a heated oven filled with NIMO. At the opposite end a 90 μm conical nozzle is mounted. The mixture of humidified buffer gas and NIMO is co-expanded through the nozzle, which leads to the formation of $NIMO(H_2O)_n$ clusters. The cluster beam is skimmed after a distance of ~2.5 cm and crossed by an electron beam in the interaction region ~1.5 m downstream. The electron energy can be varied by an accelerating potential. The created anions are extracted by a 2 μs long high-voltage pulse into a reflectron time-of-flight (RTOF) mass analyser with a mass resolution of ~$5 \times 10^3$. A delay of 0.5 μs between electron pulse and ion extraction excludes any effects caused by those. With each extraction pulse, all anions are analysed, detected by a multichannel plate and recorded as mass spectrum.

**Materials.** The NIMO used in both experiments was purchased from Toronto Research Chemicals (Canada) with a stated purity of ≥99%. For the doped water cluster study, type II pure water was prepared by reverse osmosis. Helium 4.6 and neon 5.0 served as buffer gases.

**Experimental procedures.** NIMO appears as powder but the focus lies on interactions with single molecules or molecules in a microhydration environment, the sample reservoirs of both experiments are heated to evaporate the sample. For Wippi, the oven was heated to about 95 °C, for CLUB between 80 and 110 °C. The

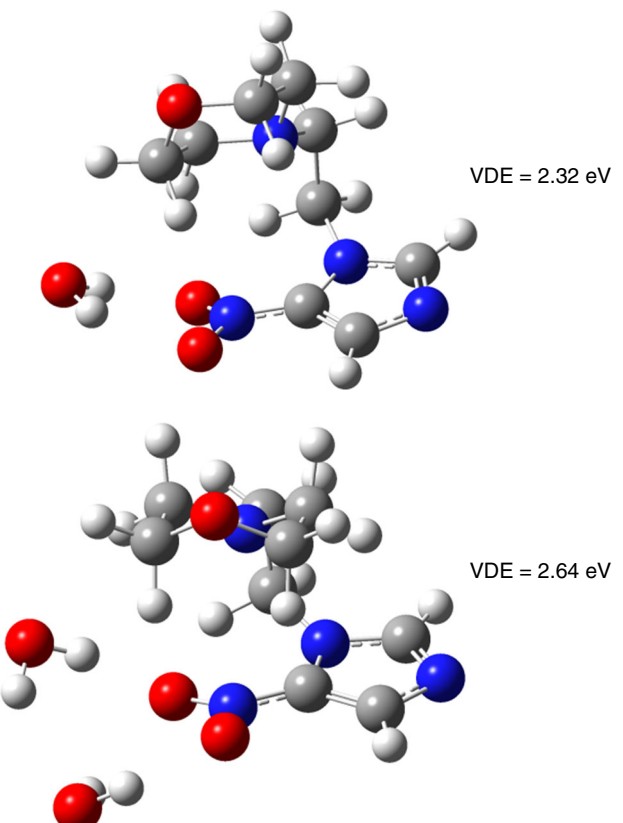

VDE = 2.32 eV

VDE = 2.64 eV

**Fig. 5** Lowest energy conformers of hydrated anionic nimorazole. The calculations were performed for one (top) and two (bottom) water molecules. The anions were optimized at M062x/6–31 + G(d,p) level of theory and basis set. The corresponding vertical detachment energy (VDE) is shown respectively. Colour coding: Carbon (grey), hydrogen (white), nitrogen (blue), oxygen (red)

temperatures were set carefully avoiding thermal decomposition of the sample but having reasonable signal rates. At CLUB, the mean cluster sizes are controlled by the pressure of the buffer gas. Here, higher pressures lead to higher humidification conditions and thus larger cluster sizes.

Additionally, background spectra were taken at identical experimental conditions to exclude both background gas and instrumentally caused peaks. For Wippi this includes both measurements with empty oven and at random masses, for CLUB measurements at empty oven and during a blocked cluster beam. Background data was subtracted from the signal traces.

For both setups, an energy calibration is required. At Wippi, the well-known 0 eV resonance positions in the ion yields of the $SF_6^-/SF_6$ [18] and $Cl^-/CCl_4$ [39] arising from s-wave electron attachment processes are used. Additionally, the energy resolution is determined based on the full-width at half maximum (FWHM) of this peak to be ~100 meV. At CLUB, electrons below 1.3 eV are strongly suppressed due to the design of the electron gun (see ref. [40] for details). Thus, the 4.4 eV resonance in the ion yield of $O^-/CO_2$ [41] is used for the calibration of the energy scale. The energy resolution amounts to ~0.7 eV. Cross-sections were determined with the data taken at Wippi by comparing the ion yields of NIMO and the well-known cross-sections of the 0.8 eV peak of $Cl^-/CCl_4$ [39] measured under the same conditions. Pressure calibrations caused by different sample introduction methods were implemented based on earlier experiments. Partial pressures were taken into account by normalizing the signal traces according to the related values. Only the order of magnitude can be derived due to systematic uncertainties. Two main influencing factors exist. First, the partial pressure determination arises uncertainties as the partial pressure cannot be measured directly but can only be evaluated by subtracting the pressures with and without presence of NIMO in the vacuum chamber. Additionally, the correction factor of the hot cathode used as pressure gauge for different gases must be estimated since it is not available (O(30%)). Second, resolution and transmission effects of the used quadrupole mass analyser result in varying signal heights (O(30%)).

**Data analysis for Figs. 1 and 2**. Statistical significance and reproducibility were verified by repeating each measurement several times (Figs. 1 and 2). In case of

Wippi data (Figs. 1 and 2), this includes data obtained at different days for various oven fillings, making up to about 200–300 set of measurements for the NIMO and nitrogen dioxide anion each, with gate times of 1 s/mass step and step size of 0.01 u. The figures in the main text show the mean of the according measurements already converted into cross-sections. Error bars refer to the statistically caused standard error of the mean which is calculated as $s/\sqrt{m}$ with $s$ the standard deviation of the mean and $m$ the number of averaged measurements. Here, the individual curves are normalised for error calculations to exclude systematic uncertainties caused by the quadrupole and pressure determination described before and hence only refer to the statistical variation of the shape of the resonance.

**Data analysis for Figs. 3 and 4**. The ratios and mean number of water molecules $<n>$ in the mixed clusters $NIMO(H_2O)_n^-$ were determined from the mass spectra depicted in the Supplementary Fig. 4 (Figs. 3 and 4). The mass spectra were obtained by averaging the values of cumulative mass spectra obtained in three independent measurements for every hydration condition except of the mass spectrum for the isolated molecule taken only once. The isolated molecule was studied in detail using the Wippi apparatus introduced above. The individual cumulative spectra used for averaging were taken at different days and different spectrometer settings. To avoid systematic errors, we performed the set of measurements at the experimental setting preferring the detection of high $m/z$'s in the present TOF setup and the standard deviation of these measurements was used for estimation of error bars in Fig. 3. Every individual cumulative mass spectrum was obtained as a sum of 21, background subtracted, and electron current divided, mass spectra measured at electron energies from 0.6 to 5.6 eV with step size 0.25 eV. The dynamic range of the single measured mass spectrum was $2 \times 10^6$.

**Quantum chemical calculations**. The thermodynamic threshold (free energy of reaction $\Delta G$) for a DEA reaction, considering precursor molecule M and a release of a neutral fragment X (see e.g., reaction (1)), can be expressed by $\Delta G([M - X]^-) = DE(M - X) - EA(M - X)$, where $DE(M - X)$ is the bond dissociation energy and $EA(M - X)$ the electron affinity of the corresponding fragment. The threshold energy for the experimental observation of $[M - X]^-$ in electron attachment experiments coincides with $\Delta G([M - X]^-)$ if the fragments are formed with no excess energy. Fragmentation reactions with kinetic energy release occur at electron energies above the thermodynamic threshold $\Delta G([M - X]^-)$. Quantum chemical calculations employing the density functional M062x[42,43] were carried out to calculate free energies of reactions, dissociation energies (including the zero-point energy correction), and adiabatic electron affinities. For NIMO we used the lowest structure reported in ref. [17]. All structures where optimized at the M062x/6–311 + G(d,p) level of theory and basis set with the Gaussian-09D01 program package[44]. Structures of NIMO hydrated by one and two water molecules were optimized at M062x/6–31 + G(d,p) level of theory and basis set. We determined AEAs and vertical electron affinities of the neutrals, and VDEs of the anions. Frequencies were calculated in all cases to confirm that the structures are local minima on the potential energy surface and not transition states.

## Data availability

All data that led to the present findings are available upon request to the corresponding author. The source data underlying Fig. 1–4 and Supplementary Figs. 1–4 are provided as a Source Data file.

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

## Acknowledgements

This work was supported by FWF, Vienna (P30332). R.M. and P.L.V. acknowledge the Portuguese National Funding Agency FCT-MCTES through PD/BD/114452/2016 and research grants UID/FIS/00068/2019 (CEFITEC) and PTDC/FIS-AQM/31281/2017. This work was also supported by Radiation Biology and Biophysics Doctoral Training Programme (RaBBiT, PD/00193/2010); UID/Multi/ 04378/2019 (UCIBIO). J.K. acknowledges the support by the Czech Science Foundation Grant 19–01159S. L.F. is grateful to the LABEX Lyon Institute of Origins (ANR-10-LABX-0066) of the Université de Lyon for its financial support within the program "Investissements d'Avenir" (ANR-11-IDEX-0007) of the French government operated by the National Research Agency (ANR). The crucial computing support from CCIN2P3 (France) is acknowledged gratefully. S.D. thanks Ass. Prof. Dr. Andreas Seppi from the Medical University Innsbruck for discussions.

## Author contributions

R.M. and J.K.: data aquisition experiment, data analysis; L.F.: quantum chemical calculations; R.M., J.K., L.F., J.F., M.F., P.L.-V., E.I. and S.D.: interpretation of data and manuscript preparation.

## Additional information

**Competing interests:** The authors declare no competing interests.

