## [Peer Review File · Nature Communications]

Reviewers' comments:

Reviewer #1 (Remarks to the Author):

Below are my review comments on "Low-energy electrons transform the nimorazole molecule into a radiosensitizer" by Stephan Denfil, et. al.

This paper reports the study of low-energy electron interactions with nimorazole and hydrated nimorazole. The results show that electron attachment occurs with an unusually high cross section and that the electron attachment process is likely important with respect to a radiosensitizer effect.

This is mostly a demonstration paper of what is already known and/or suggested regarding the use of nimorazole as a cancer radiotherapy agent. Specifically, nimorazole is already used as a standard compound for pharyngeal and supraglottic carcinoma in Danish radiotherapy centers. There are several accounts in the literature (refs. 10 and 25 of the submitted manuscript) that suggest that nimorazole only becomes active in radiotherapy after it is reduced. The observation of a stable nimorazole anion due to electron attachment of near zero eV electrons is consistent with this hypothesis.

It is also demonstrated that dissociative electron attachment (DEA) is essentially quenched due to solvation of the nimorazole; again another concept quite expected based on previous work (some of it by this group or sub-group of authors). The DEA suppression is then thought to be of minor importance especially with respect to associative attachment. Though this is true, it is also expected since the presence of water is known to lead to autodetachment of the electron. This can and is known to involve compound states...not just the nimorazole collision target. Though the autodetached electron must undergo inelastic scattering events prior to trapping on the nimorazole, these important loss channels are not addressed at all. In my opinion, it is due to the bias for stable anions in both of the measurements presented. No information regarding the neutral reactive fragments of the nimorazole or water fragments is given. Therefore, the general statement regarding DEA is a bit misleading. It is more appropriate to realize that the entrance channels for all these energy-loss channels are the formation of transient negative ions. The initial DEA decay channel which lead to stable negative ions may be of less importance in fully hydrated systems and within cells; the autodetached electrons, secondary loss channels and reactions involving reactive radicals must be considered and should not be overlooked.

In view of these issues, I cannot recommend publishing this paper in Nature. There are also less important concerns I have with this submission that the authors may wish to address.

- 1.) One order of magnitude error in the cross section is rather high for these types of measurements. The reasons for this large error should be addressed; especially since the reported value seems larger than the molecule itself.
- 2.) There are many typographical errors. For example, it is stated that the resolution was MeV. I assume the authors meant meV.

- 3.) The authors realize that many short lived products are produced. Did they attempt to measure any of these beyond the NIMO-NO₂· ?
- 4.) It is not that difficult to add a few water molecules to the complex and redo the quantum chemistry calculations to see if the barriers drop. It may only take one or two water molecules.

Reviewer #2 (Remarks to the Author):

This is an interesting paper which studies electron attachment to nimorazole (NIMO), a representative of nitroimidazolic radiosensitizers, and to its clusters with water, NIMO(H₂O)_n, in the gas phase. The comparison of single molecule experiments with microsolvation ones allowed the authors to demonstrate that solvation significantly hinders dissociative electron attachment (DEA) to NIMO which produces the NO₂⁻ anion. Indeed, for the solvated NIMO (n=14) the ion yield ratio of NO₂⁻/(NIMO(H₂O)_n)⁻ decreases by three orders of magnitude compared to the isolated NIMO molecule.

The paper describes the state-of-the-art measurements on electron attachment to single molecule and its water clusters. Also the level of theory employed to elucidate the required energetic characteristics, electron affinity and thermodynamic thresholds, is adequate. This is a very well written paper. Clear and concise. Its length seems to fit exactly to Nat. Comm. requirements.

I fully agree with author's interpretation of their experimental results. The formation of the NIMO⁻ anion radical in the cell might be, indeed, responsible for radiosensitizing action of nimorazole. However, in this point I would be a bit more cautious. The authors write: "The present results indicate that the intact anion remains as the cytotoxic radical. In a cellular environment, the compound could then bind to the DNA. For this case chemical reaction schemes exist, which then lead to strand breaks in DNA^{26,27}". However, it is not clear what reaction schemes they do mean. NIMO⁻ is an anion while DNA is a polyanion. Therefore, there is a strong repulsive potential that hinders binding of NIMO⁻ to DNA. A similar situation occurs for the reaction of solvated electron (e_{solv}) with DNA. Due to the polyanionic nature of DNA the rate constant for reaction of e_{solv} with DNA is ca. two orders of magnitude lower (Chem. Rev. 1989. 89. 503-520) than those with nucleosides (which are neutral molecules).

The paper suggests that NIMO⁻ is the key radiosensitizing species. Actually, the authors write: "Therefore, free radical formation by low-energy electrons seems to be the key process for radiosensitization by NIs". One should, however, remember about the oxygen-mimetic properties (radiosensitizing properties) of nitroimidazolic compounds. As indicated by the Wardman mechanism (ref. 10; Clinical Oncology (2007) 19: 397-417) NI does not need to attach an excess electron to lead to a strand break (SB) when SB is triggered by hydroxyl radical attachment to a

nucleobase. The authors should clarify it by emphasizing that their finding does not concern oxygen-mimetic action of NIs.

Minor remarks:

- In line 110 the authors write: "and -0.26 eV (release of C₃H₃N₂ + C₆H₁₁NO)". It is not clear why such a reaction was considered.
- Line 76: substitute "In contrast to halouracils, he dominant feature..." with "In contrast to halouracils, the dominant feature..."

Reply to the comments and suggestions of the reviewers

We would like to thank both reviewers for their reports, which helped us to improve the quality of the current work. Our detailed response and the list of corrections addressing all of the comments by the reviewers are presented below, hoping that these are to their satisfaction. We also submit the manuscript file, where all changes are highlighted in yellow.

Comments Reviewer 1

Below are my review comments on “Low-energy electrons transform the nimorazole molecule into a radiosensitiser” by Stephan Denfil, et. al.

This paper reports the study of low-energy electron interactions with nimorazole and hydrated nimorazole. The results show that electron attachment occurs with an unusually high cross section and that the electron attachment process is likely important with respect to a radiosensitizer effect.

This is mostly a demonstration paper of what is already known and/or suggested regarding the use of nimorazole as a cancer radiotherapy agent. Specifically, nimorazole is already used as a standard compound for pharyngeal and supraglottic carcinoma in Danish radiotherapy centers.

There are several accounts in the literature (refs. 10 and 25 of the submitted manuscript) that suggest that nimorazole only becomes active in radiotherapy after it is reduced. The observation of a stable nimorazole anion due to electron attachment of near zero eV electrons is consistent with this hypothesis.

It is also demonstrated that dissociative electron attachment (DEA) is essentially quenched due to solvation of the nimorazole; again another concept quite expected based on previous work (some of it by this group or sub-group of authors). The DEA suppression is then thought to be of minor importance especially with respect to associative attachment.

Answer: Briefly summarizing the criticisms in the last three paragraphs, the reviewer points out that (a) nimorazole is used as standard sensitiser, (b) it is known that it has to be reduced in order to work like that and (c) it is known that DEA is quenched upon hydration. We completely agree with the first two statements and partially agree with the third (there are exceptions, see for example Postulka et al. J. Phys. Chem. B, 2017, 121 (38), 8965–8974 and Kocisek et al. J. Phys. Chem. B, 2018, 122 (20), 5212–5217). However, none of these is the major message of our manuscript - the new finding is the extremely effective associative attachment of ballistic electrons, which is the reduction mechanism and thus a key to the radiosensitizing effect. We are not aware of this effect being considered previously.

Moreover, we pointed out in the submitted manuscript that the DEA suppression is an important aspect (see for example our statement “Together with the remarkably high electron attachment cross-section, the quenching of DEA in solvation is the second key outcome of our studies.” on page 9). It was suggested previously (see ref. 32: Edwards DI, J Antimicrob Chemoth 31, 9-20 (1993)) that in hypoxic tumor cells NO₂⁻ formation upon reduction terminates the reaction sequence to DNA damage. In the present work we can clearly demonstrate that hydration effectively quenches this process for NIMO.

Though this is true, it is also expected since the presence of water is known to lead to autodetachment of the electron. This can and is known to involve compound states...not just the nimorazole collision target. Though the autodetached electron must undergo inelastic scattering events prior to trapping on the nimorazole, these important loss channels are not addressed at all. In my opinion, it is due to the bias for stable anions in both of the measurements presented. No information regarding the neutral reactive fragments of the mimorazole or water fragments is given. Therefore, the general statement regarding DEA is a bit misleading. It is more appropriate to realize that the entrance channels for all these energy-loss channels are the formation of transient negative ions. The initial DEA decay channel which lead to stable negative ions may be of less importance in fully hydrated systems and within cells; the autodetached electrons, secondary loss channels and reactions involving reactive radicals must be considered and should not be overlooked.

Answer: We agree with the referee that secondary electrons, once formed in the cellular medium, loose in their energy in successive elastic and inelastic collisions, which also involve compound states. We mention this point now in the introduction of the revised manuscript (see page 3), since it is an important aspect in the general description of radiation damage. Beyond the introduction, a more detailed description with presentation of all possible secondary loss channels like reactions leading to neutral radicals would go beyond the scope of the present study.

However, we investigated the possibility of increased autodetachment in the NIMO compound state by the presence of water, since the associative and dissociative electron attachment processes studied here are in direct competition with autodetachment. In the revised manuscript we added the experimental data demonstrating that in the relevant electron energy range (sub-excitation energies) transient negative ion formation occurs independently of the water environment (see the related discussion added on page 7 and page 8 and new data shown in Figure 4). In addition, we carried out quantum chemical calculations for NIMO with one and two water molecules attached (see comment below), which indicated increased stability of the NIMO anion upon hydration (see page 8, new Figure 5 and new Supplementary Figs. 5 and 6).

In view of these issues, I cannot recommend publishing this paper in Nature. There are also less important concerns I have with this submission that the authors may wish to address.

1.) One order of magnitude error in the cross section is rather high for these types of measurements. The reasons for this large error should be addressed; especially since the reported value seems larger than the molecule itself.

Answer: We note that the experimental determinations of DEA cross-sections for non-volatile compounds can be a challenging task in crossed beam experiments as previously shown for the cross-section determination of the nucleobase thymine [see Kopyra et al., Int. J. Mass Spect. 281 (2009) 89–91]. One of the major problems in the course of such experiments is the condensation of evaporated molecules. Though nimorazole is less problematic in this respect, since it has a much higher vapour pressure, we decided to provide a realistic systematic error for our cross-section value and provided a detailed discussion on possible sources of errors in the Method's section (see page 12). Major error sources represent the indirect determination of the neutral beam density, and the transmission of anions through the quadrupole.

In view of the comment by the reviewer, we slightly revised the main text, by directly referring the reader to the Section Methods for a detailed discussion on the systematic error: (“...with an uncertainty of one order of magnitude, see Methods for the discussion on possible systematic errors.”, see page 4).

2.) There are many typographical errors. For example, it is stated that the resolution was Mev. I assume the authors meant meV.

Answer: We corrected the mentioned error which resulted from the non-uniform use of capital/small letters in the titles of cited articles in the reference list. Now we generally use capital letters only for the first word of the article title.

3.) The authors realize that many short lived products are produced. Did they attempt to measure any of these beyond the NIMO-NO₂· ?

Answer: We note that in the current mass spectrometric experiment we cannot directly measure the electron energy dependence of (NIMO-NO₂)[·] formation. This is only possible by the measurement of the complementary reaction product NO₂⁻. In the course of the present study, we were able to measure the ion yields of 14 weakly abundant fragment anions upon electron attachment to single NIMO. Those fragment anions were not presented in the submitted manuscript due to their low abundance. We show now the anion yields for all detected anions in the Supplementary Fig. 1-3.

4.) It is not that difficult to add a few water molecules to the complex and redo the quantum chemistry calculations to see if the barriers drop. It may only take one or two water molecules.

Answer: We carried out the calculations requested by the referee and included them in the revised manuscript (see page 8, new Figure 5 and new Supplementary Figs. 5 and 6). As mentioned above, the calculations indicate increased stability of the NIMO anion upon hydration.

Comments Reviewer 2

This is an interesting paper which studies electron attachment to nimorazole (NIMO), a representative of nitroimidazolic radiosensitizers, and to its clusters with water, $\text{NIMO}(\text{H}_2\text{O})_n$, in the gas phase. The comparison of single molecule experiments with microsolvation ones allowed the authors to demonstrate that solvation significantly hinders dissociative electron attachment (DEA) to NIMO which produces the NO_2^- anion. Indeed, for the solvated NIMO ($n=14$) the ion yield ratio of $\text{NO}_2^-/(\text{NIMO}(\text{H}_2\text{O})_n)^-$ decreases by three orders of magnitude compared to the isolated NIMO molecule.

The paper describes the state-of-the-art measurements on electron attachment to single molecule and its water clusters. Also the level of theory employed to elucidate the required energetic characteristics, electron affinity and thermodynamic thresholds, is adequate. This is a very well written paper. Clear and concise. Its length seems to fit exactly to Nat. Comm. requirements.

I fully agree with author's interpretation of their experimental results. The formation of the NIMO-anion radical in the cell might be, indeed, responsible for radiosensitizing action of nimorazole. However, in this point I would be a bit more cautious. The authors write: "The present results indicate that the intact anion remains as the cytotoxic radical. In a cellular environment, the compound could then bind to the DNA. For this case chemical reaction schemes exist, which then lead to strand breaks in DNA^{26,27}". However, it is not clear what reaction schemes they do mean. NIMO- is an anion while DNA is a polyanion. Therefore, there is a strong repulsive potential that hinders binding of NIMO- to DNA. A similar situation occurs for the reaction of solvated electron (e_{solv}) with DNA. Due to the polyanionic nature of DNA the rate constant for reaction of e_{solv} with DNA is ca. two orders of magnitude lower (Chem. Rev. 1989. 89. 503-520) than those with nucleosides (which are neutral molecules).

Answer: We agree with the reviewer that the charge state of the radiosensitiser is crucial in the interaction with cellular DNA. The submitted manuscript was misleading in this point, because we did not mention that within the relevant reaction sequence in solution, the NIMO radical anions becomes neutralized by protonation before attacking the DNA (see ref. 32 and new ref. 33: P. Wardman, Curr Med Chem, 8, 739-761). As also pointed out in the comment below, the Wardman mechanism (to which we referred previously by the sentence "For this case chemical reaction schemes exist, which then lead to strand breaks in DNA^{26,27}") relies on the action of neutral NIMO. We have clarified in the revised manuscript, that the NIMO radical anion becomes neutralized before binding to DNA (see page 9).

The paper suggests that NIMO- is the key radiosensitizing species. Actually, the authors write: "Therefore, free radical formation by low-energy electrons seems to be the key process for radiosensitization by NIs". One should, however, remember about the oxygen-mimetic properties (radiosensitizing properties) of nitroimidazolic compounds. As indicated by the Wardman mechanism ref. 10; Clinical Oncology (2007) 19: 397-417) NI does not need to attach an excess electron to lead to a strand break (SB) when SB is triggered by hydroxyl radical attachment to a nucleobase. The authors should clarify it by emphasizing that their finding does not concern oxygen-mimetic action of NIs.

Answer: We agree with the reviewer that the present results do not correspond to the oxygen-mimetic action of NIs, since Wardman proposed that neutralized NIs mimic the action of oxygen (as shown in Figure 4 in ref. 13), as mentioned in the last comment. We clearly point out now at the end of the main text that we identified the possible mechanism how NIs are accumulated in tumor cells (see page 10). This reaction step happens before subsequent protonation and binding to the DNA (see proposed reaction scheme in ref. 32). In the revised manuscript, we mention that the protonated radical anions of NIs may bind to DNA and therefore mimic the action of oxygen in oxygenated cells (see page 9).

Minor remarks:

- In line 110 the authors write: “and -0.26 eV (release of C₃H₃N₂ + C₆H₁₁NO)”. It is not clear why such a reaction was considered.

Answer: We agree with the referee that we did not explain, why we investigated also other reactions than the simple bond cleavage leading to NO₂⁻ formation. A detailed inspection of the NO₂⁻ ion yield close to zero eV revealed a weakly abundant peak structure at this energy which is not in agreement with the endothermicity of the DEA reaction accompanied by single bond cleavage. Since the simple bond cleavage reaction has a threshold of 0.53 eV, which is considerably above the experimentally obtained threshold of 0 eV, another reaction must lead to the ion yield close to zero eV. We can rule out impurity signals and therefore we calculated the threshold energies for more complex dissociation processes with different combinations of two neutrals. All reactions investigated showed a threshold well above 1 eV, except the mentioned one which was the only exothermic reaction.

The NO₂⁻ ion yield close to zero eV was not visible in the intensity map shown in Figure 3 of the submitted manuscript due to its low abundance. In the revised manuscript, the NO₂⁻ ion yield is plotted in the Supplementary Figure 3 with a scale, where also low abundant features are visible. In addition, we revised the text of the manuscript on page 6, and explain why we carried out more extensive calculations.

- Line 76: substitute “In contrast to halouracils, the dominant feature...” with “In contrast to halouracils, the dominant feature...”

Answer: We revised accordingly.

Other revisions:

Since the previous manuscript was directly transferred from a submission to Nature, we adapted the style of the revised manuscript to that of a manuscript for Nature Communication. This implies that we provide now an unreferenced abstract and subsection headings. In addition, the supplementary information is provided now in a separated file.

REVIEWERS' COMMENTS:

Reviewer #2 (Remarks to the Author):

The authors satisfactorily responded to all my critical remarks and corrected the manuscript accordingly. Now, the paper can be published as it is.

Reply to the comment of the reviewer #2

Comment: The authors satisfactorily responded to all my critical remarks and corrected the manuscript accordingly. Now, the paper can be published as it is.

Answer: We would like to thank reviewer #2 for reading the revised manuscript again and for his favourable reports.

Editorial note:

Reviewer #1 was unable to assess the revised version of this work. Instead, Reviewer #2 provided the Editor with comments on the authors' responses to Reviewer #1's concerns.